# Delayed-mode reprocessing of in situ sea level data for the Copernicus Marine Service

Jue Lin-Ye[1,*], Begoña Pérez-Gómez[2], Alejandro Gallardo[1], Fernando Manzano[2], Marta de Alfonso[2], Elizabeth Bradshaw[3], Angela Hibbert[3]

[1] Nologin Ocean Weather Systems. Paseo de la Castellana 216, Planta 8ª, Oficina 811, 28046 Madrid, España

[2] Puertos del Estado. Avenida del Partenón 10, 28042, Madrid, España

[3] National Oceanography Centre. Joseph Proudman Building, 6 Brownlow Street, Liverpool L3 5DA, United Kingdom

[*] *Correspondence to*: Jue Lin-Ye (jue.lin@nologin.es)

**Abstract.** The number of tide gauges providing coastal sea level data has significantly increased in recent decades. They help in the issue of coastal hazard warnings, the forecasting (indirectly, through models) of storm surges and tsunamis, and in operational oceanography applications. This data is automatically quality controlled in near-real-time, in the Copernicus Marine Service. A new initiative seeks to provide delayed-mode reprocessed data for the Copernicus Marine Service, by developing a new product and upgrading the software used in its automated quality control. Several new modules, such as buddy checking or detection of attenuated data, are implemented. The new product was launched in November of 2022. The entire reprocessing is discussed, in detail. An example of the information that can be extracted from the delayed-mode reprocessed product is also given.

Keywords: sea level, in situ, delayed-mode, software

## 1 Introduction

Sea level has long been in the public spotlight for being one major indicator of climate change. It is now an essential ocean variable[1]. Remote and in situ (non-remote) observations show that the sea level rise has been accelerating in recent decades (Nerem et al., 2018; Dangendorf et al., 2019). It raises concerns about a corresponding increase in the magnitude, frequency and impact of coastal extreme events (Vousdoukas et al., 2018; Kirezci et al., 2020; Almar et al., 2021). Tide gauge data are an essential source of information to understand the contribution of different physical processes to sea level variations, at different timescales. Therefore, reliable delayed-mode reprocessed sea level data are crucial for validation of hindcasting and forecasting ocean models, for both short-term forecasts and long-term climatic studies.

In the past, the quality control of sea level measurements heavily relied on expert visual inspection, mainly done at national level by tide gauge network operators. This is still a key part of the process, but the community has increasingly adopted automation to detect low quality data, more quickly and exhaustively (IOC, 2020), both in near-real time (NRT) and delayed mode, depending on the application. Likewise, expert knowledge, ruled by physical formulation, can be partially translated into machine language. The combination of automation and an efficient algorithm can minimize the total cost of the quality control, and can make it possible to apply quality control to a large network of thousands of tide gauge stations. Up to now, automatic quality control was applied in NRT for in situ sea level data in the Copernicus Marine In Situ Thematic Assembly Centre (In Situ TAC). However, a more comprehensive and exhaustive quality control in delayed mode, including additional tests and validation methods of historical

timeseries, was not performed, at a centralized site, for generation of a higher quality dataset in this service (Sea Level Reprocessed or Multi-Year product). This is essential to ensure a more reliable use of these data for scientific applications, including aggregation into existing datasets that download data from the In Situ TAC (Eurosea project. Deliverable 3.3), and it is the main achievement of the new product described in this manuscript, especially for those hundreds of stations where this reprocessing and validation of the historical timeseries is not available or done at national level.

**2 The Copernicus Marine Service, the In Situ Thematic Centre (In Situ TAC) and the data dealt with in the quality control**

The Copernicus Marine Service provides regular and systematic reference information on the physical and biogeochemical ocean and sea-ice state for the global ocean and the European region seas (Le Traon, 2019; Copernicus Marine In Situ Tac Data Management Team, 2021). Four key application areas of the Copernicus Marine Service are:

- maritime safety
- marine resources
- marine and coastal environment
- weather, seasonal forecasting and climate

Observations are used by Copernicus Marine Service Thematic Assembly Centres (TACs) to provide high-level data products, as well as by Copernicus Marine Service Monitoring and Forecasting Centres (MFCs) to validate and constrain their global and regional ocean analysis and forecasting systems. The sources for the data can be satellites or in situ observations.

The Copernicus Marine Service In Situ Thematic Assembly Center (In Situ TAC) is the main interface between the Copernicus Marine Service and the global, regional and coastal in situ observing networks. It collects, processes and carries out the quality control of the upstream in situ data, required to both constrain and directly validate modeling and data assimilation systems, and to directly serve downstream applications and services. The in situ network includes the network of tide gauges, argos, research vessels, moorings, gliders, surface drifters, among other methods of measurement. The in situ observations are for sea level, wave, temperature, salinity, currents, chlorophyll, oxygen, nutrients, pH and fugacity of $CO_2$. The In Situ TAC delivers NRT products, that are delivered within 24 hours, having completed automatic quality processing, as well as scientifically assessed reprocessed products.

Until November of 2022, the tide gauges only provided NRT data product, while a sea level delayed-mode reprocessed product was pending. This functionality is also unavailable, until this date, in other major databases, such as the Global Extreme Sea Level Analysis (GESLA, which aggregates existing NRT data in In Situ TAC) and the Global Sea Level Observing System (GLOSS), which rely on national providers efforts on quality control and processing. There is, currently, a contract under way to deliver this delayed-mode reprocessed sea level product for the Copernicus Marine Service (Product User Manual, 2022; Quality Information Document, 2022). A first release was made in November of 2022. Its product and the software used to carry out the quality control at that moment is described herein. The spatial coverage consists mostly of European waters, at this moment (Fig. 1). The most veteran tide gauges are in the Baltic and the Iberian-Biscay-Irish regions. Some of these stations still operate, and offer series of over 100 years.

The time series used in this product span very different periods, ranging from 1886 in some Baltic stations, to dozens of stations that were installed as late as 2019 (Fig. 2). The newer sensors have different precisions, which tend to be in millimeters, compared to older sensors, used at some point in the century-long series, that can have a precision of centimeters. Following the Baltic area, more tide gauges were successively installed in the Iberian-Biscay-Irish region, the Mediterranean region, the Arctic, the North-West shelf, and more recently, the Black Sea region (Fig. 2). Most NRT data in the Mediterranean Basin had series shorter than one year, as of 2020. New tide gauges are being installed in Turkey and other Eastern Mediterranean countries. Some geographically non-European islands also contribute, at the global level, to this product.

**3 The NRT: The NRT (L1) product and the SELENE software**

The NRT process (Fig. 3) in Copernicus takes total sea levels at the original time sampling (as measured by the tide gauges), applies an automatic quality control test, and flags sea level data points, where appropriate. These series are transformed into an intermediate, 5-min sampled format, where short gaps (below or equal to 27 minutes) are interpolated. Then, a Pugh filter (Pugh, 1987) is applied to produce hourly sea levels. In the Copernicus Marine Service, the original, automatically flagged total sea level and the hourly filtered total sea level are both available as NRT products.

The quality control of tide gauge time series is performed with the SEa LEvel NEar-real time quality control processing (SELENE) software (Puertos del Estado, 2019). The original software was created in the 1990s, based on Fortran77 programming language and C-Shell scripts. It included quality control, filtering to hourly values and computation of non-tidal residuals. The NRT service (every 15 minutes) in the Copernicus Marine Service now uses the SELENE software adapted to Python language.

The SELENE software follows best practices, as defined by GLOSS (IOC, 2020). In NRT, it applies the L1 quality control level, as defined by the IOC (2020), including: out-of-range, spike detection, and stability tests. The quality control of SELENE software uses a series of station-configuration parameters. These are the maximum/minimum total/de-tided sea levels; the degree of the polynomial for spike detection, as well as the size of the window of time to be fit to the polynomial, and the maximum standard deviation allowed for the non-spike data; and the limit for the stuck-data (data that keeps being constant for an abnormally long time) to start to constitute one (that is, once the data is constant for as long as the limit time, it is considered to be stuck).

Prior to the launch of the NRT operational service, a sensitivity test was carried out on tide gauges from Puertos del Estado. The data in the test had a sampling of 1 minute. The polynomial used for total sea levels was optimal when being of 2 degrees; the best window size was of 200 points (sampling points) and the maximum standard deviation was equal to 4. The polynomial used for de-tided sea levels was optimal as of 3 degrees; the best window size was of 500 points and the maximum standard deviation was equal to 5. The limit for stuck-data was equal to 10 points. The limits for the maximum and minimum total sea levels and the de-tided sea levels were given by an expert, for each tide gauge. This set of criteria was extended to the Iberian-Biscay-Irish region in the NRT automatised quality control in the Copernicus Marine Service.

The original de-tiding module of SELENE is based on the Foreman tidal prediction package in Fortran77 (Foreman, 1977). The de-tided sea level observations facilitate quality control, since tidal variations often dominate a sea level record, making subtle errors more difficult to detect. On the contrary, bad data can easily present itself as abrupt changes in the non-tidal residual, while being told apart from the real extreme surges, which evolve more gradually. The corrected time series were then used to improve the sea level forecasts of the Nivmar Sea Level Forecasting system of Puertos del Estado (Álvarez-Fanjul et al, 2001; Pérez-Gómez et al., 2013). This was performed by means of a nudging technique, still operational today. However, the de-tiding module is only available in internal reprocessing in Puertos del Estado, but is not yet applied in operational NRT mode in the Copernicus Marine Service in situ NRT product, as it requires the computation of yearly tidal constants at each station in the Copernicus Marine Service.

The interest of international tide gauge communities (such as the Copernicus Marine Service, GLOSS and EuroGOOS Tide Gauge Task Team) was for the SELENE software, first used in the Copernicus Marine Service as the NRT quality control of total sea level, to be a modern, open-source (Python) and well-documented format. It had to be multi-platform (Linux, MacOS and Windows) and it is now used by some regions in the Copernicus Marine Service In Situ TAC, and by some national providers for purposes outside the Copernicus system.

**4 The delayed-mode: the reprocessed product (L2) and the upgraded SELENE software**

The delayed-mode reprocessing follows a similar procedure as the NRT process (Fig. 3). The products (https://doi.org/10.48670/moi-00307) available in the first release of November 2022 comprise quality-controlled versions of the original series of total sea level and the hourly filtered series of sea level. These products end in December

2020, and are available for the 639 stations that were operational by that time. The series that, by December of 2020, presented less than one year of data, were excluded (Fig. 1). The reprocessed product has been assigned with corresponding metadata, to better identify the product from each station. An updated version of the SELENE software, developed in 2022, is used for the automated quality control. The reprocessing is complemented by visual inspection. The reprocessing can more accurately flag the bad or dubious data than the NRT product, which can only rely on automated quality control of a short moving time window (a few days), applied every 15 minutes. With the new tests implemented for delayed mode in the first release of this product, the reprocessing flags 1.9% of the sea level data as bad/dubious data, whereas the NRT automated quality control flagged about 0.3% of the sea level measurements as bad/dubious data. In some tide gauges, the total sea level is already reprocessed by national providers. In this case, the data mode is read from the source metadata of each sea level file, and recorded as a station-configuration parameter. When the data mode is "reprocessed", the original flagging is assumed to be correct, although some further automated and visual quality control is carried out, by default.

The upgraded SELENE software contains a series of new modules. The changes can, in their turn, improve the NRT quality control. A new version of the de-tiding module had been incorporated in Python language. The corresponding Python package is called uTide (Codiga, 2011), also based upon Foreman. It is used by other authors, such as Mélet et al. (2021), for tide gauge data reprocessing. uTide accepts non-uniform sampling frequencies. This flexibility comes in especially handy, since some time series uploaded to Copernicus Marine Service are not entirely uniform, in terms of sampling interval. Astronomical constants are given for each year which large gaps (>0.5 week) add up to less than 1 month. Also,

$$\log_{10}\frac{max\left(A\right)}{min\left(A\right)} \leq 0.7$$

for all months, where A is the monthly amplitude of the M2 tidal component. If a year does not comply with these conditions and astronomical constants are not computed for that year, the de-tiding process can borrow the constants from up to three years previous or later than that year for its computation.

Hourly de-tided sea level and astronomical tide for the Iberian-Biscay-Irish region will be available as delayed-mode reprocessed products, in the Copernicus Marine Service, by November 2023. Neither of them contain the non-astronomical "tide". That is, cyclical fluctuation of the sea level that are generated by non-astronomical phenomena. The de-tided sea level can, internally, help pinpoint anomalous behaviour, such as in Fig. 4. These small errors are not easily detectable by visual inspection, and this automation can significantly reduce workload. Other issues detectable with de-tided sea levels are clock malfunction and datum changes, when combined with visual inspection.

A new module added to the SELENE software is the "buddy" checking, applied to hourly de-tided sea levels and monthly mean sea levels. It has been implemented to compare the target station to neighboring stations within a range of 0.01° (approximately 1 km). Another module is the detection of attenuated data. The attenuated data is easily detectable by a human, but it is difficult to have a machine tell it apart from neap tides. Neap tides are more centred around the the mean water level of the sea level, and attenuation usually presents itself as a fluctuation around a value far away from the mean water level. Specific testing had been carried out, to set the parameters and the criteria to tell the two cases apart. Because of the level of complication of the specific criteria, the reader may refer to the Product User Manual.

Finally, a sensitivity test was carried out, to calibrate the station-configuration parameters used by SELENE, on the tide gauges in the Copernicus Marine Service. A unified criterion for station-configuration parameters is created. It is designed to be further used on other regions of the planet. The limits for maximum and minimum total sea levels are the mean water level plus or minus the 99[th] percentile of the total sea level in the whole time series. The limits for maximum and minimum de-tided sea levels is the same as for the total sea level, but the mean water level is set to be null, as it is eliminated from the data during de-tiding. The polynomial, used to detect spikes in the total sea level, is of 2 degrees; the maximum standard deviation allowed for non-spikes is equal to 3. The window size of total sea level data to be fitted to the polynomial depends on the frequency of sampling. The greater the sampling frequency, the more the elements that should be within the window (see Product User Manual for details). The window for fitting de-tided sea levels is double that of the window for total sea levels. This is because the total sea level has cyclicities caused by astronomical tides, whereas the de-tided sea level does not have such cyclicity. These customized parameters can improve the performance of SELENE by successfully detecting spikes, even in difficult cases (Fig. 5).

The limit for stuck-data starts with 10 points. Then, the limit can be more lax in the case of reduced precision (in
centimeters, instead of millimeters) or when the data is flagged as "stuck" in over 50% of the total time series. This
situation is commonplace for older tide gauges. If, despite the modifications to the limit of stuck-data, the total sea level
is still being flagged often during a visual inspection, the time series can be classified as having "many stuck" or "too
many stuck". In this case, the limit can be further modified, depending on the sampling frequency (Product User Manual).

The processing speed of the first Python version of the SELENE software (currently used in the NRT service) was slower
than that of the original Fortran77 code in Puertos del Estado. The software was only computationally possible on a time
window of 15 minutes, the one used for NRT quality control. However, the reprocessing of extensive series in a delayed-
mode reprocessed product was very costly. In the upgrade of the SELENE software, the computational cost of the Python-
based algorithm has been reduced ten-fold. One change is that, while the version of SELENE used in the NRT process
dealt with every time $t$ to find spikes, the delayed-mode reprocessing starts the detection of spikes only in specific cases.
The increment of data at the time $t$ must be larger than 100 millimeters, and greater than the following value

$$\frac{1}{50} \frac{\left| slev_{max} - slev_{min} \right|}{3},$$

where $slev_{max}$ and $slev_{min}$ are the maximum and minimum total sea levels.

**5 Possible applications**

The reprocessed total sea level product has many applications for scientists and coastal stakeholders. As an example,
seasonal extreme water levels can be computed, normalized (divided) by the tidal range. Seasonal relative maximum
sea levels derived in this way could help highlight regions at most risk of flooding. Alternatively, seasonal relative
lowest water levels can also be useful to remind harbours to keep their maintenance on schedule, to ensure their
channels have enough draft for ships to circulate freely.

From hourly (filtered) total sea level, the maximum and minimum values for each season are computed. The values for
all stations are scaled, to share a single mean water level. These extreme sea level parameters are subsequently divided
by the tidal range at each station, in order to produce the relative extreme sea levels. The tidal range is computed as the
difference between the 99[th] percentile and the 1[st] percentile of the whole time series. The relative maximum and
minimum hourly total sea levels in each season are shown in Figs. 6 and 7, respectively.  Note that the used series are
the ones that by December of 2020 presented a time interval longer than one year. Moreover, each series has a different
length, as shown in Fig. 1.

According to the European Environment Agency (eea.europe.eu) and to the reprocessed sea level data, the (absolute)
highest water levels are in the Iberian-Biscay-Irish area and the Northwest-shelf. The areas with the most extreme water
levels are the British coasts in the Irish Sea, as well as the French coast in the English Channel, both near the Atlantic
Ocean. Some high-water levels are typical in the Frisian Sea, near the German-Denmark border. This area, too, is
heavily influenced by the Atlantic Sea. However, the Iberian-Biscay-Irish area and the Northwest-shelf are
geographically and technologically more prepared for such ranges of sea level, since they are typically macro-tidal.
Attention must be paid to micro-tidal and meso-tidal areas with large percentage increases in sea level, for small
changes can induce floods in them.

The Strait of Kattegat (between the east coast of Denmark and the South-western coast of Sweden, located at 56.82ºN,
11.39ºE) exhibits the highest relative sea levels in Spring, Autumn and Winter (Fig. 6). Furthermore, during the
Autumn-Winter, the phenomenon of high relative sea levels extends to the neighboring Gulf of Bothnia (the gulf
between Sweden and Finland, located at 62.33ºN, 19.55ºE). The Strait of Kattegat also presents the lowest relative sea
levels in Autumn and Winter (see Fig. 7). The North-western Mediterranean displays high relative sea levels in Spring,
Autumn and Winter. It is known that some of these regions, like Barcelona, in the North-western Mediterranean, have a
history of engineering challenges for coastal protection, like during Storm Gloria (Pérez Gómez et al., 2021). Similarly,
Denmark and Sweden had taken steps to improve their protection against floods, because of the historical flood in 1872
(Hallin et al., 2021; Fredriksson et al., 2018). The upgrades were not always based on engineering, but also on changes

in the structure of the economic system, insurance policies, among others. Much more information can be drawn from the reprocessed sea level data. All this will improve communication between stakeholders.

**6 Conclusions**

The Copernicus delayed-mode reprocessed sea level product is crucial for validation of ocean models, for both short-term forecasts and long-term climatic studies (reanalysis).  The quality of the data can determine both the veracity of the conclusions and the correctness of the decisions to be taken. The Copernicus Marine System already has a service of operational automated quality control of NRT data, which detects spikes, stuck data, among other errors. It uses a software called SELENE.

The delayed-mode reprocessed product described here was first available in November of 2022, and was obtained by applying an upgraded version of SELENE software, in delayed mode, including additional tests and  visual inspection of the whole time series by an expert in sea level data. The upgrade includes an adaptation of the de-tiding module from Fortran77 to Python language, the implementation of the buddy checking of hourly de-tided and monthly mean sea levels, the detection of attenuated data, and the creation of a criterion for determining the station-configuration parameters used in the quality control. The criterion was obtained after carrying out sensitivity tests on each module of the SELENE software, such as the spike detection test and the stuck-data test, intended to be applied to the whole planet. The upgrade to SELENE has increased the ability of automatically flagging bad data by 1.6%, as compared to the automated quality control in the NRT process. The computational cost of the upgraded software is also 10 times lower, if comparing the shared modules. This faster speed is possible due to, in part, the pre-selection of data to apply spike detection.

The new delayed-mode sea level product is of major interest for the scientific community and other data aggregators using In Situ TAC sea level data, and more broadly for coastal and local decision-makers, such as coastal engineers intensively using tide gauges data to design coastal and port infrastructures. It is also an added-value product for those regions or countries where quality control of historical records is not performed at national level.  Product enhancements are planned, such as improvement of the geographical coverage, by including stations from the GLOSS network in 2024. Time extensions of the product to six months prior to the respective release will be regularly provided. Future goals for the reprocessing service include to compare monthly mean sea levels to altimetry, as well as a feasibility study on the inclusion of vertical land motion corrections, based on Global Navigation Satellite System (GNSS) receivers.

**Competing interests**

The contact author has declared that none of the authors has any competing interests.

**Acknowledgements**

We use the uTide python package in the upgraded SELENE software. Many thanks to the developers for their help solving questions raised during the implementation of the tool.

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

N. years

**Fig. 1 Time series length of tide gauges that are included in the reprocessed product, first release of November of 2022.**

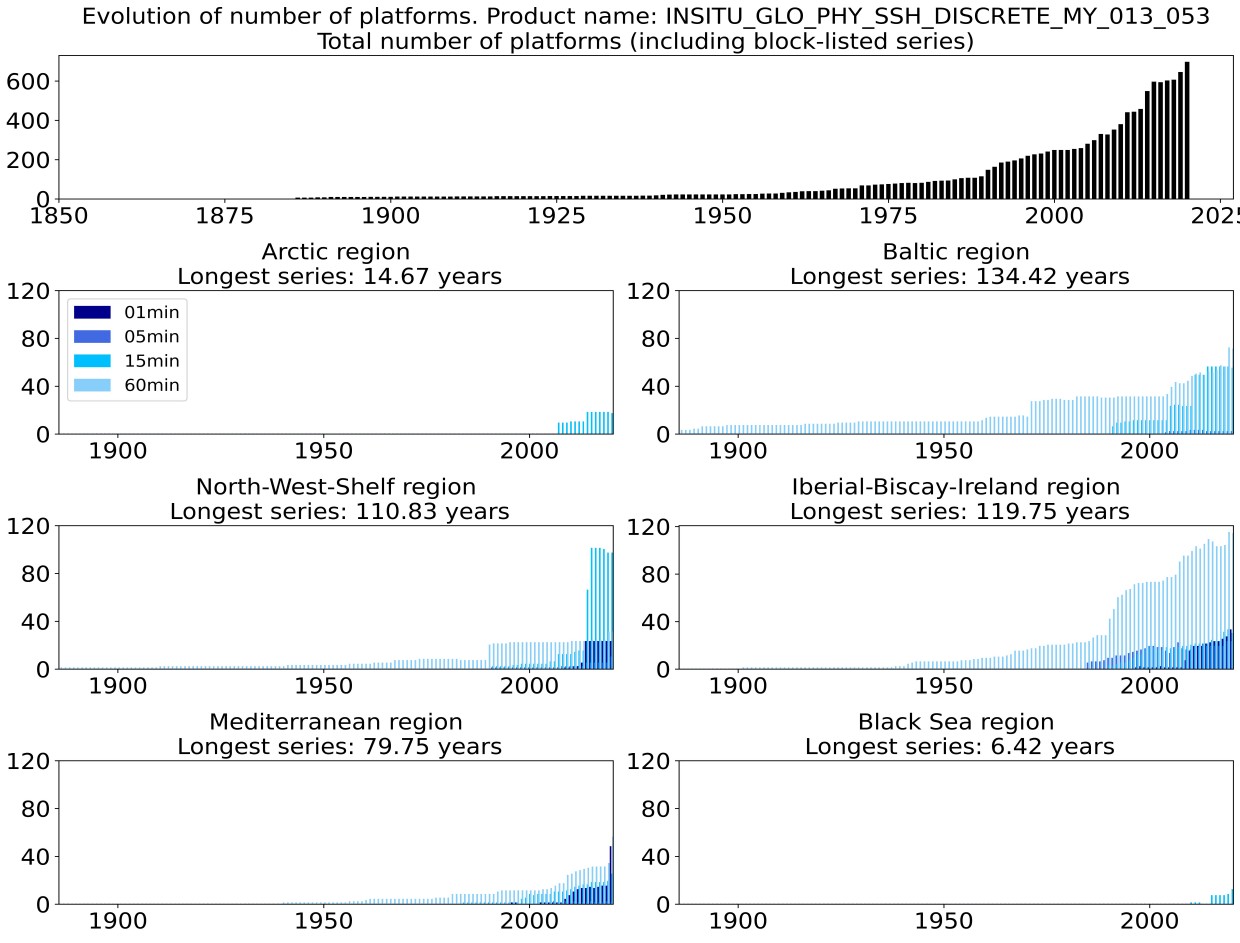

**Fig. 2 Evolution of the number of tide gauges per year in Copernicus Marine Service In Situ TAC. The figure**
**includes the series that span less than one year at the end of 2020. Product name:**
**INSITU_GLO_PHY_SSH_DISCRETE_MY_013_053.**

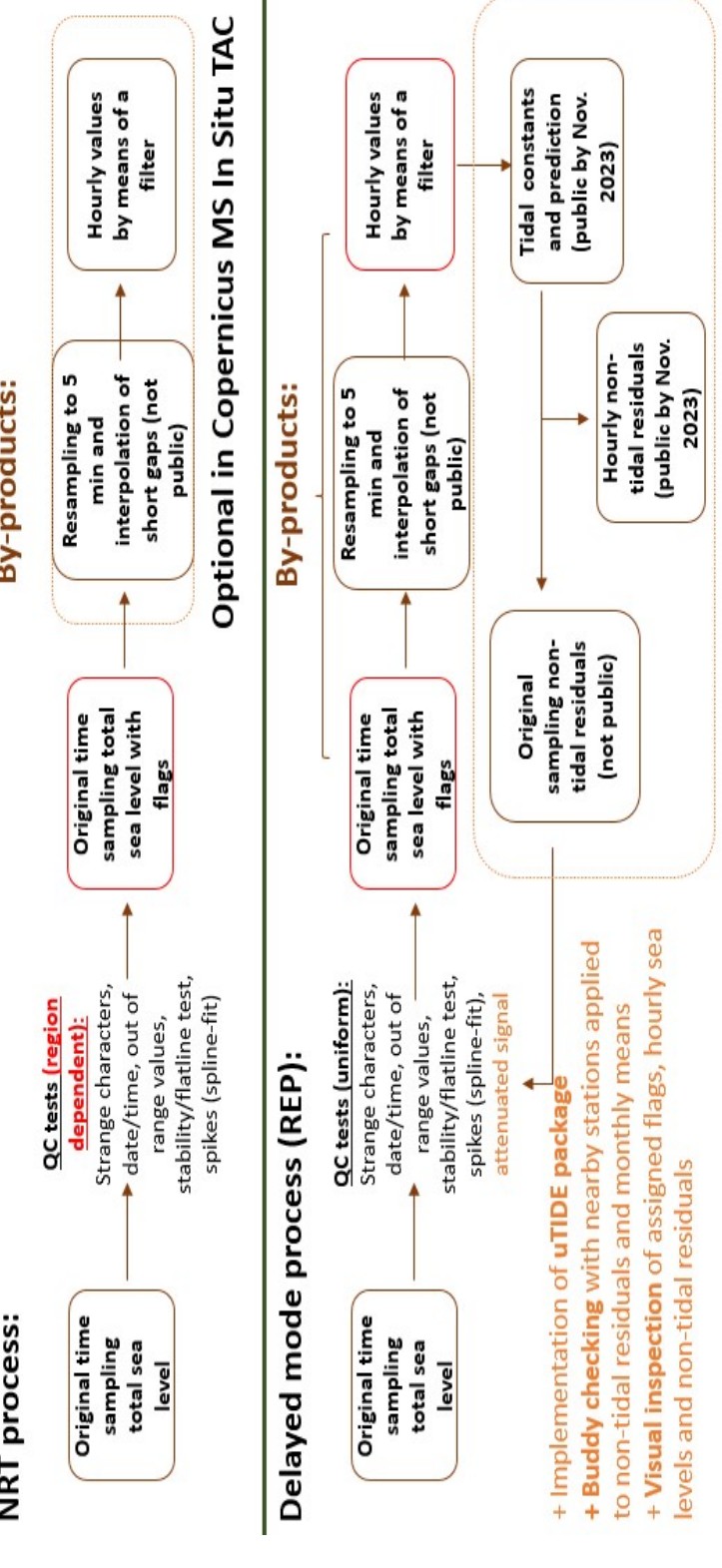

**Fig. 3 Flow chart of NRT (near-real-time) process and delayed-mode reprocessing.**

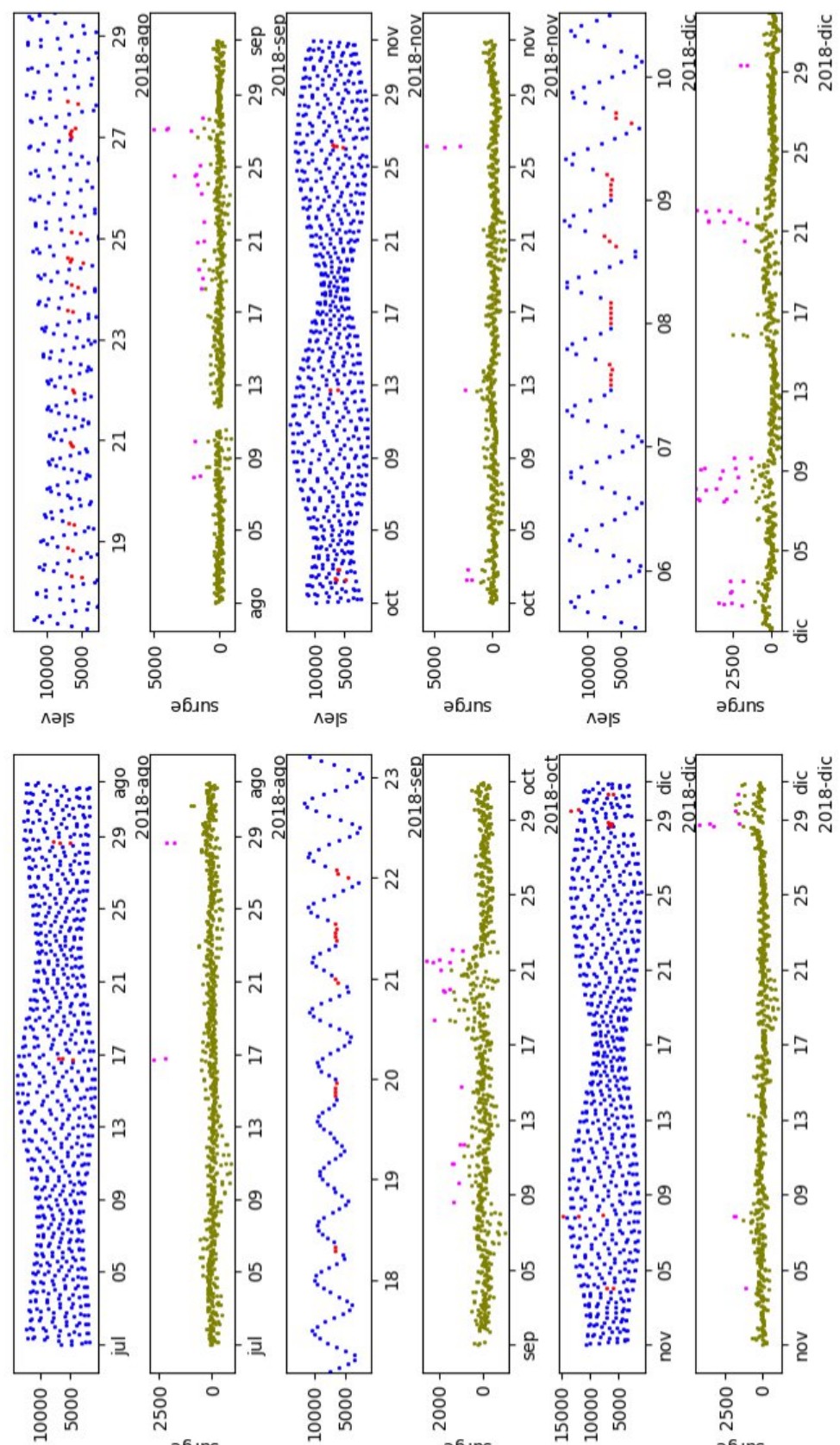

**Fig. 4 Example of detection of bad data in the total sea level ("slev", in millimeters), with the help of de-tided sea level ("surge", in millimeters).
Subplots for the total sea level, in August, September and December are zoomed in to see the detection of bad data. Bad data are shown in red in the
total sea level, and in pink in the de-tided sea level.**

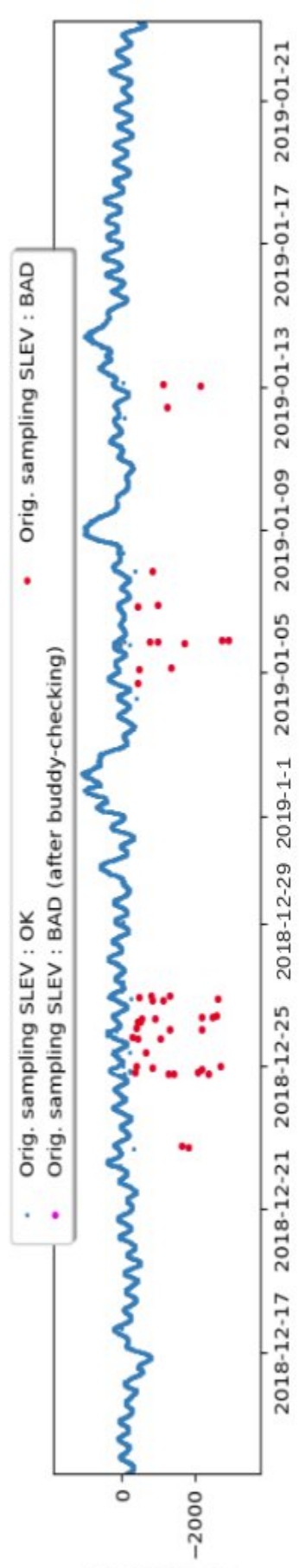

**Fig. 5 Example of spike detection in the total sea level**

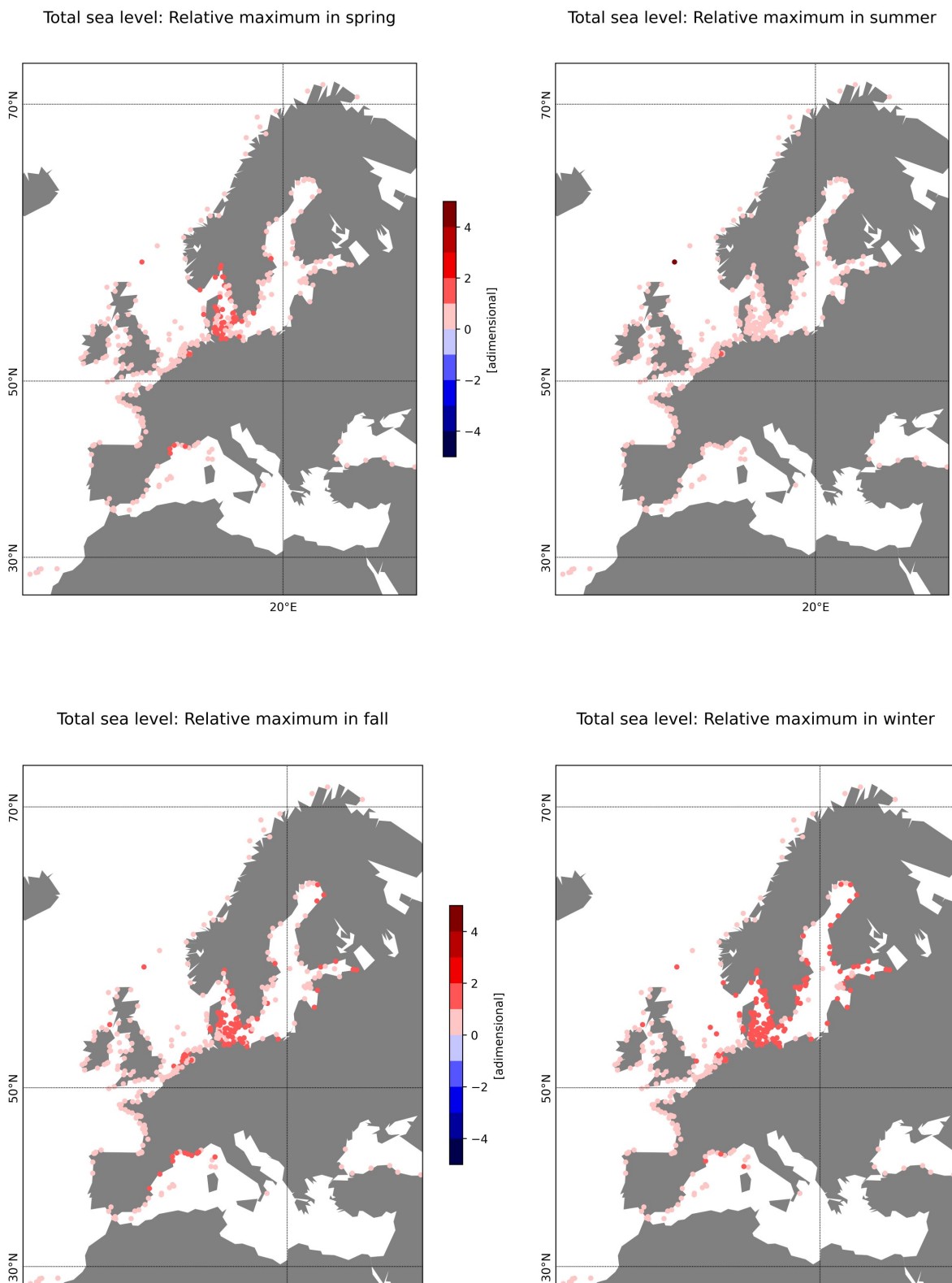

1 **Fig. 6 Relative maximum sea levels (maximum sea levels at each station, divided by the corresponding**
2 **tidal range), in each season.**

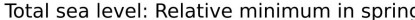
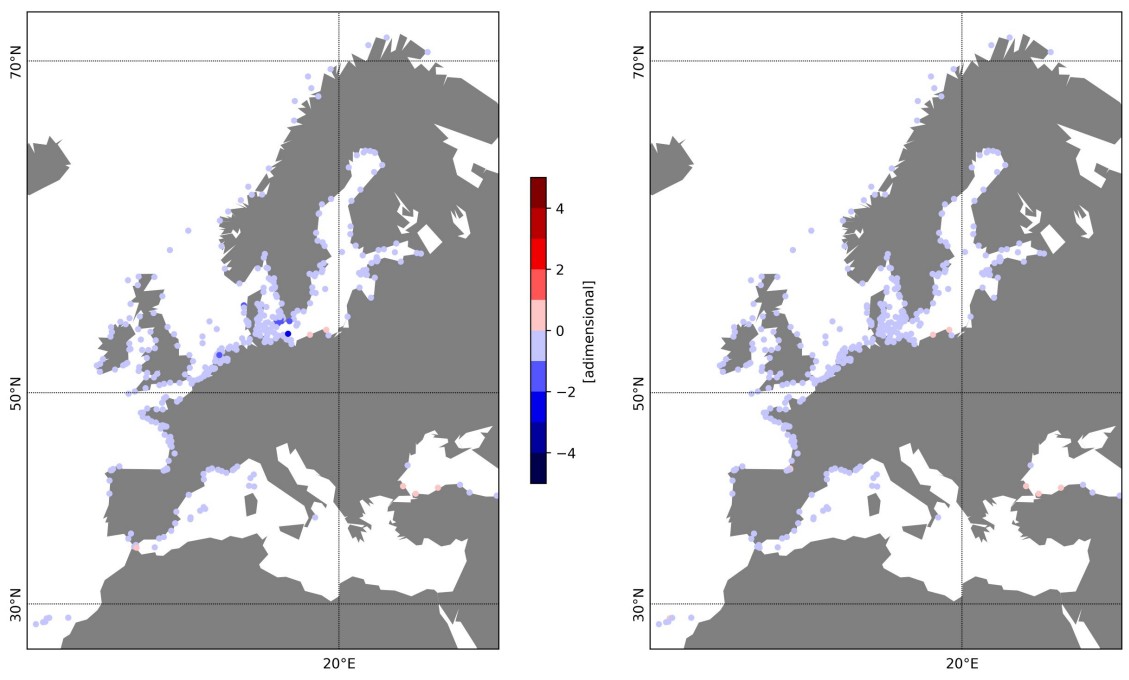
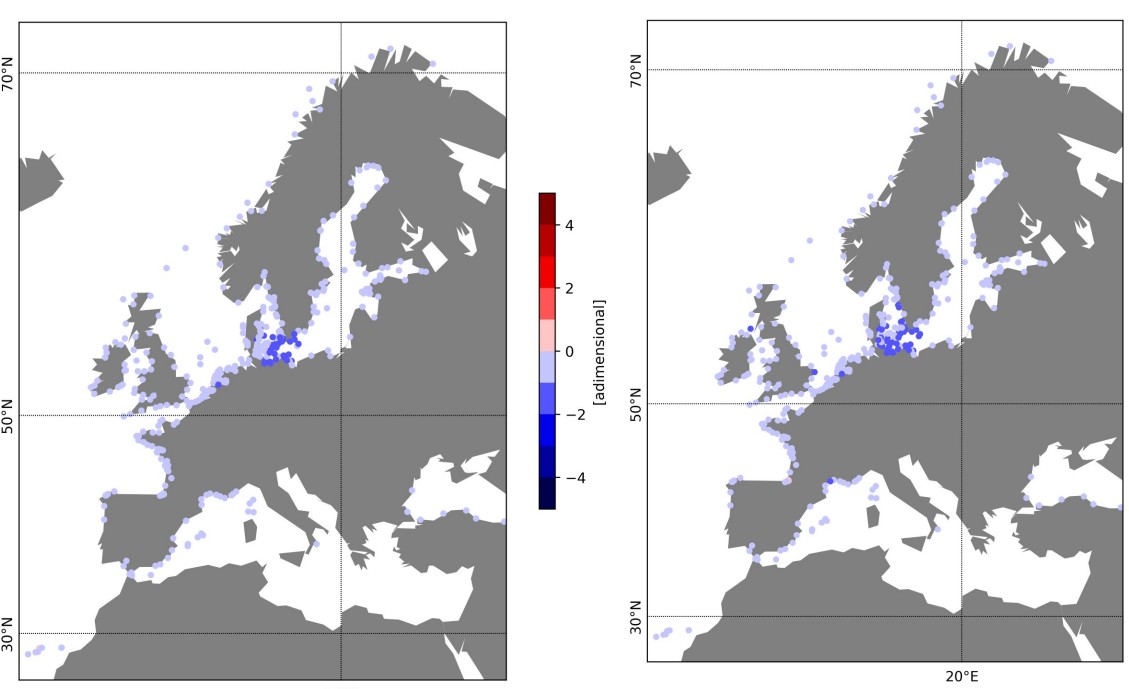

1    **Fig. 7 Relative minimum sea levels, in each season.**

