# Peer review of "Delayed-mode reprocessing of in situ sea level data for the"

_EGUsphere, 2023_

## Author Response (AR1)

**Answer to reviewer #1:**

**Summary**

**This manuscript presents a new re-processing product for tide gauge data.**

*First of all, we would like to thank the reviewer for the time and dedication to provide with these reasoned comments.*

**Major comments**

**To be honest, I found this a quite difficult paper to review. To me, it reads more like a deliverable from a project then as a scientific publication: there are references to contracts, manuals and deliverables, while the explanation of what is actually done, what are the scientific advances, and how does this new method improve previous products is relatively limited.**

*We have rewritten the manuscript, thoroughly. What is actually done is described in Section 4 (Section 3, in the previous version). The advances would be the various new modules, such as buddy checking. The new modules, as well as the comment on the improvement compared to near-real-products (see below), should constitute the overall enhancements over previous products.*

**For starters, the manuscript doesn't explain what delayed-mode reprocessing is – quite crucial to get all readers along.**

*Please see answer to P1 – L18, below.*

**It also doesn't mention other tide gauge databases/products and how this new dataset relate to other existing ones such as PSMSL, GESLA and GLOSS.**

*We have added the following comment (p 42 – L36-39): "This functionality is also unavailable, up to date, in other major databases, such as the ones owned by the Global Extreme Sea Level Analysis (GESLA, which aggregates existing NRT data in In Situ TAC) and the Global Sea Level Observing System (GLOSS), which rely on national providers effort on quality control and processing. "*

**One thing that might help also is to improve the figures and the discussion thereof, most labels are really small and the figure captions are very short, and I think these should be expanded to aid the reader and explain what is actually shown in the figures.**

*Please see answers to the figures, below.*

**When a figure goes into more scientific analyses (figs 7/8), the data shown and reasoning why this particular example is chosen, is not very clear.**

*The example given is just one of many possible uses that could be given for this product, for scientific purposes. It is intended, to show how quality controlled data can be greatly helpful. This might seem quite obvious, but non-quality controlled data would hamper the conclusions given in the text, thus distorting them.*

**Something I'd for instance be interested to see is how does near-realtime compare to delayed-mode: are there large differences, what is the added value?**

*We have added the following comment on page 3: "The reprocessing can more accurately flag the bad or dubious data than the near-real-time product, which can only rely on automated quality control of a short period of time (15 minutes). The reprocessing flags 1.9% of the sea level data as bad/dubious data, whereas the near-real-time automated quality control can flag about 0.3% of the sea level measurements as bad/dubious data. "*

**Minor comments**

**P1**

**The abstract lacks clarity, for instance**

**L15-16 - why is this required?**

*We have quality control of the near-real-time data that is given by the providers (every 24hours), but we need to, in hindsight, carry out a quality control on the totality time series of sea levels.*

**L 16 – 'from these platforms'; not clear which platforms are referred to here?**

*We have substituted "from these platforms" by "from the tidal gauges",
which is equivalent and is clearer in meaning.*

**L18 – can you explain what 'delayed-mode reprocessed' means?**

*Indeed, but for the sake of fluidity, we prefer to leave the word as
is in the text. It is explained in section 3 that "delayed-mode" is a
L2 product. It is the ocean measurement, in this case, a sea level,
that has been further gone under a quality control.*

**L19 – 'visually controlled' – what does this mean?**

*To be 'visually controlled' means to be quality controlled by a human
expert. We intend to leave this expression as is, in the text.*

**P2**

**L10-11 - 'tide gauges are emerging' sounds a bit odd. 'are being
installed/set up' or 'new tide gauge data is retrieved/available'?**

*It is true that it sounds somewhat odd. We have taken the
recommendation and changed "emerging" for "being installed".*

**Figures general: In addition to the captions (see above) the figures
are not very colour-blind friendly and tend to have small text labels.**

*Colors in figures have been turned into monochrome in different
shades, so people with any kind of color blindness can see the
gradation. It is also true that text labels are small, which has to do
with the overall size of the figure. Therefore, the images have been
re-sized.*

**Fig 1; 'global' dots are difficult to see, can the symbols be
larger?**

*We have changed Fig. 1 by Fig. 3.*

**Fig 5: red and pink are difficult to distinguish; axis labels are
very small. Perhaps mention which stations these are?**

*Fig. 5: There is no need to discern between red and pink. Red is for
flagging bad data in total sea level. Pink is for flagging bad data in
surge. We have enlarged the axis labels, along with the figure, as*

*well. We do not mention the name of the stations, as we intend for them to be anonymous, in order to avoid conflicts of interest.*

**Fig 6; figure text is very small. Which station is this?**

*Fig. 6: We have enlarged the figure text, along with the figure. We neither mention the station in this figure, in order to avoid conflicts of interest.*

**Fig 7/8; increase symbol sizes?**

*These figures are intended to show colors, rather than individual tide gauges. Therefore, we would like to keep the symbol size.*

**Could the authors please explain what is shown as it is very puzzling how the 'relative maximum/minimum' is non-dimensional? Also from reading the main text (p4) it doesn't become very clear what the meaning is of these 'relative maxima/minima' – how to interpret them, why is this shown in this way?**

*In order to explain what the "relative extreme sea levels" are, we have extended one sentence, so now it reads "These extreme sea level parameters are subsequently divided by the tidal range at each station, in order to produce the relative extreme sea levels. "*

**On p4, l30-36 talk about 'absolute' water levels, why are these not shown?**

*We specify in the text that " according to the European Environment Agency and to the reprocessed sea level data".*

**Grammatical; please check uses of 'it' and if possible replace them - it is not always clear to what 'it' refers to exactly (for instance p1- L 16, L30, L32; fig2 caption)**

*We rewritten the text, so the pronoun 'it' only appears when it is explicit. In some other instances, we use some other word, or the specific word that it refers to, in order to increase clarity. In some other cases (p1- L30, p2- L33, p4- L44 ), 'it' appears when referring to a non-personal verb.*

———————————

**Answer to reviewer #2 (Laurent Testut)**

*First of all, we would like to thank Dr. Laurent Testut for such valuable comments, based on his expertise on this field.*

**General comments**

**The main objective of this article is to present the reprocessing strategy used by the Copernicus Marine Service to compute a new delayed-mode sea level product based on the reanalysis of many tide gauges timeseries. This objective is of major interest for the scientific community and more broadly for many coastal engineers using intensively tide gauges data to design coastal defences for example. The number of stations included in the new produt is very important (#639) and I guess the effort made to achieve this reprocessing is huge. However, the presentation of the work certainly detracts from the quality of the underlying work.**

*Indeed, sir, we have rewritten the letter, in order to reflect all the accomplished work behind the service.*

**Many sentences are approximatives,**

*We have deleted some approximate sentences, such as (P1): "As one of the cheapest, most readily monitored ocean variables, it is of no surprise that it was one of the first ones to be systematically measured."*

**the quality of the Figure is often poor**

*We have modified the figures, to improve their quality.*

**and the abstract/introduction and conclusion quite below what one would expect from a scientific publication.**

*We have rewritten the abstract and the introduction.*

**I really recommend the authors to concentrate on the core of the paper by (i) presenting a detailed version of the process**

*We have rewritten Sections 2 and 3 (changed to Sections 3 and 4, in the current version of the manuscript), which detail the near-real-time and the reprocessing processes.*

**and (ii) demonstrated the added-value of such a product (which I guess should be quite easy).**

*In Section 4, we try to demonstrate the added value of the reprocessed product, which is our main focus in this paper. This section is also thoroughly re-written.*

**If this article focuses on the data quality control and its contribution, especially in comparison with real-time, then this work would certainly make a good publication.**

*Indeed, we have put great effort into improving Section 4 for this version of the manuscript, in order to explain the upgrading in quality control, in the reprocessing product, in comparison with the near-real-time product.*

**Specific comments**

**The Abstract/Introduction is full of approximate statements that should be cleared off the text (see some examples in the next section "Technical corrections"). I encourage the co-authors of this preprint to help in the rewriting of a more concise et precise Abstract/Introduction/Conclusion.**

*The abstract, the introduction are the conclusion have been rewritten, in order to meet the standard set by the reviewer.*

**The presentation of the Copernicus Marine Service from P2, L1 to P2, L13 (and P2, L21 to L26) is very interesting. It can be extracted from the introduction, extended a bit and put into a dedicated section that briefly present CMEMS, the In situ TAC and the main data product on which is applied the QC.**

*We have created a section (Section 2) to talk about the Copernicus Marine Service, the IN SITU TAC and the main data product (the near-real-time product) on which we applied the QC.*

**The actual section 2 and 3 on the presentation of the real-time and delayed-mode are the core of the paper. They are easy to follow, but the section on the delayed-mode could be extended to show the real added value of the delayed-mode QC.**

*We have rewritten the sections about the near-real-time product that is used for the QC (and the SELENE software used there, which is the*

*same one used in the delayed-mode product) and the delayed-model product (and the upgraded SELENE product).*

**More examples of the capability of the processing to flag wrong data could be interesting to see. Comparison between the delayed-mode and real-time is also important to discuss, to cearly see in which case the delayed-mode is able to correct what real-time have missed.**

*We have rewritten Sections 3 and 4, in order to clarify these issues.*

**What is the % (or number) of values that is flagged by the real-time and by the delayed-mode (in the worst and best case, or the mean stats).**

*We have added the lines (Section 4, page 4, L16-20): " The reprocessing can more accurately flag the bad or dubious data than the NRT product, which can only rely on automated quality control of a short moving time window (a few days), applied every 15 minutes. With the new tests implemented for delayed mode in the first release of this product, the reprocessing flags 1.9% of the sea level data as bad/dubious data, whereas the NRT automated quality control can flag about 0.3% of the sea level measurements as bad/dubious data. "*

**The last section on the possible application, does not in my opinion add any value to the paper. I'm sure it is necessary at all to have this section on "possible application". A sentence or two in the conclusion should be enough to convince the reader of the interest of this new product. The author should concentrate on the quality of the strategy they use to reprocess the data and on the added value of this new product.**

*We would like to keep this section, anyway, to prove some scientific value to the reprocessed data, despite being obvious.*

**Technical corrections**

**P1, L13-14 : TG can helps in triggering the warning system but not really sure that it helps in forecasting the tsunamis.**

*The aid shall be indirect, by data assimilation into forecasting models. We add the comment "indirectly, through models", in the Abstract.*

**P1, L15 : When you speak about the "historical sea level timeseries" of the Copernicus system, I guess you are talking the "archived sea level timeseries". In the sea level community "historical sea level" can be confused with very long sea level records, which is not you are talking about I guess.**

*We actually mean both "very long sea level records" and short records. We refer to the series, from now on, as just the "whole series".*

**P1, L32-33 : (i) I'm not sure it is convenient to define ocean variable by their price**

*We have rewritten the Introduction, to avoid mentioning the prices.*

**(ii) I pretty sure that many other ocean variable are more easy and cheap to measure than sea level (ie temperature)**

*We have rewritten the Introduction.*

**(iii) atmospheric parameters were systematically recorded way before sea level. This sentence is a typical example of an "approximate" statement that is present a lot in the introduction.**

*We have rewritten the introduction.*

**P2, L14-20 : This section on the physical processes affecting the sea level is particularly weird in between the presentation of the CMEMS system. I suggest to remove this section.**

*We have removed it and suppose it to be obvious.*

**P2, L44-45: remove "control of the total sea level time series" in the sentence**

*Done.*

**P3, L5 : How do you deal with the maximum surge event ? Is the maximum total sea level can be wrongly flag in case of extreme surge ?**

*We added the sentence (P3, L39-41): "On the contrary, bad data can easily present itself as abrupt changes in the non-tidal residual,*

while being told apart from the real extreme surges, which evolve more gradually. "

**P3, L18 : I don't understand the meaning of the last sentence, please develop.**

*We changed the word "study" for "calibration". Maybe it is clearer, now. The calibration of the parameters for the quality control is described in the next section. Therefore, we keep this sentence of Section 3 simple. (The sentences has been moved in the rewritting of the section)*

**P3, L41 : Give more explanation about the new module dealing with the attenuation of the data.**

*We added the sentences (P5 L1-6): " Another module is the detection of attenuated data. The attenuated data is easily detectable by a human, but it is difficult to have a machine tell it apart from neap tides. Neap tides are more centred around the the mean water level of the sea level, and attenuation usually presents itself as a fluctuation around a value far away from the mean water level. Specific testing had been carried out, to set the parameters and the criteria to tell the two cases apart. Because of the level of complication of the specific criteria, the reader may refer to the Product User Manual. "*

**P4, L10 : from ". Produc enhancement ..." this should be moved to the conclusion part of the paper.**

*Done.*

**Figure 1 : I'm not sure this global is really necessary**

*We have eliminated it and have used Fig. 3 to show the study area.*

**Figure 2 : The image resolution is not good on the pdf.**

*We have modified the figure, accordingly.*

**Figure 3 : Not easy to see the useful information on this global map. An histograms of the timeseries length could be more informative than a map.**

*We intend for this image to provide more insight of the geographical distribution of the longest running stations. We are worried that a histogram would be less informative for more international readers.*

**Figure 4 : ok**

—

**Figure 5 : There is a problem in the x-axis of sum of the subplots that are not corresponding. The most obvious case is the two right bottom subplots.**

*We have added the following comment to the caption: "Subplots for the total sea level, in August, September and December are zoomed in to see the detection of bad data."*

**Figure 6 and > : Not needed in my opinion.**

*Figures 6 and 7 (Figures 7 and 8 in the previous version) are linked to Section 5 (Application, it was section 4 in the previous version). Please see answer above.*

---

## Author Response (AR2)

**Reviewer #1**

**Thank you to the authors for taking my suggestions on board.**

 **A few minor things remain, but these are mostly technical corrections.**

 **Abstract**
 **L16 I'd suggest to replace 'such as buddy checking, detection of  attenuated data, among others,' by 'such as buddy checking and detection  of attenuated data,'.**
*Done*

 **Introduction**
 **L23 Climate Change = climate change**
*Done*

 **Section 2**
 **L8; the data dealt 'with' in the quality control?**

*Done*

 **L17 'Thematic'**

*Done*

 **L20 'in situ observations'**

*Done*

 **L29 NRT data products**

*We call these observations that have gone through an automated quality control a "product". We call the reprocessed observations "products", as well.*

 **L30 up to date = for now, or, until this date?**

*Done*

 **L30 'such as the Global Extreme..' (remove 'the ones owned by')**

*Done*

**L32 rely = relies, effort = efforts**

*"Rely" means that both databases "rely". We have changedd "effort" to "efforts".*

**L39 ranging = starting ?**

*It is ranging from 1886 to 2019.*

**L41 remove 'factual'?**

*Done*

**L43 'the north-west European shelf'? though this suggests that tide gauges are placed in the sea. 'The north-west European coast'?**

*The IN SITU TAC classifies the region as the "north-west European shelf", despite it actually being the coast. It is not fully accurate, but it is better identified with this name.*

**Section 4**
**L8 can flag = flagged?**

*Done*

**L11 considered = assumed?**

*Done*

**L30-36 = If the buddy checking only considers stations within 1 km of each other, how many stations does this affect? It will not be a lot of stations where this module is used?**

*Sometimes we do not find any eligible neighbor to do the buddy checking.*

**L1 worse precision = reduced precision?**

*Done*

**L7 'this was only sustainable' – can you please rephrase, not clear what this means?**

*(Section 4, L7-9) We have repharsed: " The software was only computationally possible on a time window of 15 minutes, the one used for NRT quality control. However, the reprocessing of extensive series in a delayed-mode reprocessed product was very costly."*

**L10 'tackled all time t' – can you please rephrase, not clear what this means?**

*(Section 4, L11-12) We have rephrased: "One change is that, while the version of SELENE used in the NRT process dealt with every time t to find spikes, the delayed-mode reprocessing*

*starts the detection of spikes only in specific cases. "*

**Section 5**
**L29 'according to the EEA' – can you provide a reference?**

*We proceed to reference to the website, listed in the References.*

**L32 Atlantic Ocean?**

*(Section 5, L32) We have completed the sentence: "The areas with the most extreme water levels are the British coasts in the Irish Sea, as well as the French coast in the English Channel, both near the Atlantic Ocean. "*

**Section 6**
**L12 the mentioned upgrade = the upgrade**

*Done*
* * *
*Reviewer #2*

**General comments :**
 **-----------------**
 **The main objective of this article is to present the reprocessing  strategy used by the Copernicus Marine Service to compute a new  delayed-mode sea level product based on the reanalysis of many tide  gauges timeseries. This objective is of major interest for the scientific community and more broadly for many coastal engineers using  intensively tide gauges data to design coastal defences for example. The  number of stations included in the new produt is very important (#639)  and the effort made to achieve this reprocessing is huge. The revised  version of the manuscript has taken into account many of the comments  made and this version has been improved. The Abstract/Introduction reads  much better now, with a clear and synthetic introduction and the  presentation of the Copernicus Marine Service in section 2 is clear and comprehensive. The section 3 and 4 are informative about the different  processing used by the Copernicus Marine Service. Section 4 give a  simple example of application for this dataset. Although this article  isn't strictly speaking a scientific paper, but rather a briefing note  on a new product of interest to the marine level community, I suppose it deserves to be published in a journal with a wide readership.**

 **Technical corrections :**
 **-----------------------**
 **P2, L17 : replace Thmatic by Thematic**

*Done.*

**P8, L3: There is a pb with the Pugh et al reference**

*Corrected.*